# Photosynthetic Toxicity of Enrofloxacin on *Scenedesmus obliquus* in an Aquatic Environment

**DOI:** 10.3390/ijerph19095545

**Published:** 2022-05-03

**Authors:** Zhiheng Li, Xianghong Zhang, Hong Fang, Xuanyu Lin, Xinmi Dai, Huijun Liu

**Affiliations:** 1School of Environmental Science and Engineering, Zhejiang Gongshang University, Hangzhou 310018, China; 21514028@zju.edu.cn (Z.L.); zxh2391054546@163.com (X.Z.); eliuakf@163.com (H.F.); lxy1200020315@163.com (X.L.); zjgsu_xxh@163.com (X.D.); 2Instrumental Analysis Center, Zhejiang Gongshang University, Hangzhou 310018, China

**Keywords:** aquatic environment, antibiotics, *Scenedesmus obliquus*, photosynthetic toxicity

## Abstract

Aquaculture facilities are a potential source of antibiotics in aquatic environments, having adverse effects on the algae species. In this study, the toxicity induced by enrofloxacin (ENR) on the algae *Scenedesmus obliquus* was evaluated. The uptake of ENR and the change in the growth and photosynthesis of algae were analyzed. At the exposure doses of 10–300 μg/L, the accumulated levels of ENR in algae were 10.61–18.22 μg/g and 12.09–18.34 μg/g after 48 h and 96 h of treatment, respectively. ENR inhibited the growth of algae, with a concentration for 50% effect of 119.74 μg/L, 53.09 μg/L, 64.37 μg/L, and 52.64 μg/L after 24 h, 48 h, 72 h and 96 h of treatment, respectively, indicating the self-protection and repair ability of algae in a short period of time. Furthermore, the chlorophyll contents decreased in all treatment groups, and the photosynthetic system Ⅱ parameters decreased in a dose-dependent manner under ENR stress, suggesting that ENR caused a disorder in the electron transport of the photosynthesis of algae, and the carbon fixation and assimilation processes were thus damaged. These results indicate that ENR poses a considerable risk to aquatic environments, affects the carbon sinks, and even has an adverse effect on human health.

## 1. Introduction

Rapid population growth and economic development have led to the emission of a massive amount of carbon-containing substances into the environment in recent decades, resulting in global warming. Aquatic environments, such as coastal zones, wetlands, offshore waters, and deep oceans are the Earth’s main carbon sinks [1,2]. Blue carbon is stored as biological biomass and sediment [3,4]. As a basic type of biomass, algae are able to conduct photosynthesis, synthetize organic matter, and play an important role in carbon storage in aquatic environments [5,6]. For instance, lightly or moderately calcareous algae could help seagrasses store sedimentary carbon and greatly improve blue carbon sequestration [7,8]. Moreover, as the main producer in freshwater, algae are essential in providing energy to higher living organisms. Many algae are rich in proteins, vitamins, and minerals and contain essential nutrients, such as phycobilin, that are used in food, medicines, and cosmetics [9,10]. However, due to their single-cell nature, algae are vulnerable to toxicological compounds, and their self-protection ability is thus limited compared with terrestrial plants [11,12]. In this case, evaluating the activity of algae under adverse environments is necessary and meaningful for human health.

To meet the high protein requirements of the rapidly growing population, animal production is continuing to expand globally. To decrease the risk of disease transmission in livestock, antibiotics have been widely used as disease prevention and treatment agents [13,14,15,16]. It has been estimated that the use of antibiotics in animal feed will increase by 67% compared with 2010 to 105,596 tons in 2030 [17]. The use of manure has led to an increase in antibiotic residues that can be released into rivers, lakes, groundwater, and even marine environments through surface runoff and soil infiltration. Antibiotics have been detected in aquatic systems at ng/L to μg/L levels across the globe [14,18,19]. Wei et al. detected a large amount of quinolone antibiotics in wastewater from a large aquaculture plant at concentrations of 10.3–211 μg/L [20], which pose a huge risk to aquatic organisms, especially algae. Thus, we urgently need to study the toxic effects of quinolone antibiotics on the growth and activities of algae.

Studies show that many algae are highly sensitive to antibiotics [6,8,19]. As a type of organic pollutant, antibiotics can inhibit the growth and reproduction of algae by different mechanisms due to the diverse and complex molecular structures [21,22,23]. Current studies mainly focus on the effects of antibiotics on the growth inhibition and physiological index of algae. Nevertheless, these studies ignore the accumulation of antibiotics in algae, which is closely related to food safety via the food chain. Meanwhile, as an important source of nutrients and energy, photosynthesis is the basis upon which algae obtain organic substances and O_2_ to support normal growth [24,25,26]. Previous research found that photosynthesis plays an important role in resisting environmental stresses by accelerating and promoting the Calvin cycle [27]. However, the effects of quinolone antibiotics on the photosynthesis of algae remain unclear, and little is known about the accumulation of antibiotics in algae. Therefore, exploring the effects of antibiotics on the photosynthesis system of algae is sure to help us uncover the toxicological mechanism of antibiotics to hydrophytes and elucidate the influencing factors of blue carbon sequestration.

As a typical quinolone antibiotic, enrofloxacin (ENR) is widely used as an animal medicine due to the superior antibacterial effects on nearly all bacteria and excellent tissue distribution [24,28]. To shed light on the aquatic toxicity mechanism, we determined the growth inhibition, chlorophyll content, and chlorophyll fluorescence parameters of photosynthetic system II (PS II) of *S. obliquus* induced by ENR under different exposure concentrations and durations. The distribution of ENR in the aquatic system was also explored to elucidate the adsorption and accumulation of ENR in algae. The present study unravels the toxicity mechanism of ENR on algae and provides a new evaluation of the mechanism by which quinolone antibiotics reduce the carbon sink in aquatic systems.

## 2. Materials and Methods

### 2.1. Chemical and Reagents

Enrofloxacin (ENR) with a purity of ≥98% was purchased from J&K Scientific (Beijing, China). Other chemicals and reagents were obtained from Sinopharm Chemical Reagent Co., Ltd. (Shanghai, China) and Sigma Aldrich (Shanghai, China) and were all of analytical grade.

### 2.2. Microalgal Cultures

The *S. obliquus* used in this study was obtained from the Freshwater Algae Culture Collection at the Institute of Hydrobiology (FACHB, Wuhan, China). The *S. obliquus* was cultured in HB liquid medium according to the guideline of the Organization for Economic Cooperation and Development (OECD). A total of 5 mL of pre-cultured algal cells during the period of logarithmic growth were transferred into Erlenmeyer flasks containing 90 mL of HB liquid medium (Beijing Solarbio Science & Technology Co., Ltd. Beijing, China), and ENR exposure solutions (5 mL) were then added to the test flasks in triplicate to make the final concentrations of ENR in the algal growth inhibition tests 10, 20, 50, 80, 120, 150, 180, 200, 250, and 300 μg/L. The test flasks were randomly placed in a growth incubator for 4 days under the pre-culture conditions. Detailed information about the composition of the HB medium can be found in Appendix A.

The *S. obliquus* was inoculated in sterile HB culture medium at the ratio of 1:9. The bottle was sealed with four layers of gauze before it was put into the incubator. The cultivation conditions were as follows: temperature, 25 ± 1.0 °C; light intensity, 4500–5000 lux; photosynthetically active radiation (PAR), 63 μmol·m^−2^/s; light/dark cycle, 12 h/12 h. The culture process was carried out continuously and statically, and the bottles were shaken 3–5 times a day manually to ensure that the algae were evenly distributed in the culture medium, had fully absorbed the nutrients, were not floating or sinking, and had not adhered to the wall. The algae were transferred into another clean bottle to produce the next generation when the algal density reached the logarithmic growth period. 

### 2.3. The Growth Inhibition of S. obliquus under Antibiotic Treatment

#### 2.3.1. Determination of Algal Density

The *S. obliquus* during the period of logarithmic growth was diluted by a series of gradient solutions, and the cells were then counted directly using an optical microscope to determine the density of the algae cells according to the standard method of aquatic toxicity experiments [29]. Simultaneously, the absorbance of the algal liquid was measured by a spectrophotometer at the wavelength of 680 nm, and the relationship between the cell density and the absorbance of *S. obliquus* was then obtained.

#### 2.3.2. Determination of the Growth Inhibition Rate and EC_50_

A stock solution of ENR was prepared in double-distilled water (ddH_2_O) and then filtered through a membrane with a pore size of 0.22 μm (Shanghai TITAN Technology Co., Ltd., shanghai, China). The exposure solutions of ENR were freshly prepared before each experiment. The exposure concentrations in the study were set to 10, 20, 50, 80, 120, 150, 180, 200, 250, and 300 μg/L according to the preliminary experiments. This range covers the concentrations used for ENR in aquaculture. The group without the ENR treatment was also included as a control.

All the experimental facilities were sterilized before the ENR treatment. A total of 10 mL of algal cells during the period of logarithmic growth was mixed with 90 mL of HB liquid medium thoroughly, and the bottles were sealed with four layers of gauze and placed into a light incubator for 96 h. The culture conditions were the same as described in Section 2.2. The absorbance of the algal cells was measured using a spectrophotometer at 680 nm after 24 h, 48 h, 72 h, and 96 h of culture. The biomass of the algal cells was calculated using the relationship between the algal cell density and the absorbance obtained from Section 2.3.1. The growth inhibition rate (GIR) was calculated using Equation 1, and the concentration for 50% effect (EC_50_) was calculated by fitting the Logistic model [30].
(1)$$GIR\;\left(\%\right)=\frac{{Χ}_{0}-Xn}{X_{0}}\times 100\%$$
where *X*_0_ and *Xn* indicate the algal cell density of the control and treatment groups, respectively.

### 2.4. Measurement of Photosynthetic Parameters of S. obliquus

#### 2.4.1. Determination of Chlorophyll Content

A total of 40 mL of algae treated with ENR for 48 h and 96 h was centrifuged at 1500× *g* for 10 min at 24 °C, and the supernatant was discarded. A total of 10 mL of ddH_2_O was then added to the residue, which was further centrifuged for 10 min, and the supernatant was then discarded. A total of 10 mL of anhydrous methanol was finally added and the algae was placed into a 60 °C water bath for 20 min before centrifugation for 10 min. This step was conducted in the dark. The OD_645_ and OD_663_ of the supernatant were then measured by a UV-vis spectrophotometer. The method for calculating the chlorophyll content was as follows:(2)$$\mathrm{Chl}\;a\;\left(\mathrm{mg}/L\right)=12.7\;\mathrm{OD}663-2.69\;\mathrm{OD}645$$
(3)$$\mathrm{Chl}\;b\;\left(\mathrm{mg}/L\right)=22.9\;\mathrm{OD}645-4.68\;\mathrm{OD}645$$
(4)$$\mathrm{Total}\;\mathrm{Chl}\;\left(\frac{\mathrm{mg}}{L}\right)=20.2\;\mathrm{OD}645+8.02\;\mathrm{OD}663$$
where Chl a, Chl b, and Total Chl indicate the chlorophyll a, chlorophyll b, and total chlorophyll content, respectively.

#### 2.4.2. Determination of Chlorophyll Fluorescence

A total of 30 mL of algae treated with ENR for 4 d was centrifuged at 1500× *g* for 10 min at 24 °C. A total of 1 mL of ddH_2_O was then added to the residue, and the resulting solution was mixed thoroughly for 10 s. The culture samples were measured after 20 min in the dark. The fluorescence parameters were calculated using the following equations:(5)$$F_{v}=F_{m}-F_{0}$$
(6)$$F_{v}/F_{0}=\frac{F_{m}-F_{0}}{F_{0}}$$
(7)$$F_{v}/F_{m}\;=\frac{F_{m}-F_{0}}{F_{m}}$$
where F_m_, F_0_, and F_v_ indicate the maximum fluorescence, minimum fluorescence, and variable fluorescence, respectively. Equations (6) and (7) represent the maximum photochemical quantum yield of PS II.

The interval of intense light emission was set to 20 s, and the maximum amount of photochemical light was emitted in the order of 0, 1, 36, 81, 146, 231, 336, 461, 611, and 801 μmol·m^−2^/s. *Y* (II), NPQ, *Y* (NO), and ETR were set according to the change in PAR, and they were calculated as follows: *Y* (II) = △F/F_m_’(8)
NPQ = F_m_/F_m_’−1(9)
*Y* (NO) = 1/(NPQ+1+qL(F_m_/F_0_−1))(10)
(11)$$\mathrm{ETR}=Y\left(\mathrm{II}\right)\times \mathrm{PAR}\times 0.84\times 0.5$$
where *Y* (II), NPQ, *Y* (NO), and ETR indicate the actual photosynthetic efficiency, the non-photochemical quenching coefficient, the quantum yield of non-regulatory energy dissipation, and the relative electron transfer efficiency, respectively.

### 2.5. Determination of ENR Content

The culture solution was collected after 96 h of treatment and centrifuged at 12,000× *g* to obtain the supernatant. The concentration of ENR in the solution was measured after filtering with a 0.22 μm membrane (Tianjin Jintang Experimental Equipment Co., Ltd. Tianjin, China). 

Next, the collected precipitant obtained from the previous step was ground evenly in a 10 mL solution of acetic acid: acetonitrile (*v:v* = 4:96), ultrasonicated for 5 min, vortexed for 1 min, and centrifuged at 8000× *g* for 10 min, and finally the supernatant was collected. The above procedure was repeated 3 times to obtain 30 mL of supernatant. We concentrated the supernatant at 55 °C to nearly dry with a rotary evaporator, collected the concentrated liquid, and added 5 mL, 2 mL, and 2 mL of acidified acetonitrile to the residue successively to obtain the concentrated liquid using a rotary evaporator. A total of 10 mL of n-hexane was used to redissolve the solution, and we retained the acetonitrile layer (the lower layer) but discarded the n-hexane layer (the upper layer). This step was repeated 3 times to obtain the extract. Finally, the extracted liquid was blown using nitrogen flow to nearly dry, the residue was redissolved in 1 mL of acetonitrile and filtered with a 0.22 μm membrane, and the concentration of ENR in the *S. obliquus* was determined.

The concentrations of ENR in the solution and in the lysate of *S. obliquus* were determined using high-performance liquid chromatography (HPLC, Agilent 1200, Santa Clara, CA, USA). An FLD fluorescence detector was selected; the excitation and emission wavelengths were 278 mm and 453 mm, respectively; and the parameters of the C18 chromatographic column were 150 mm × 4.6 mm × 5 μm (ANPEL Laboratory Technologies Inc., Shanghai, China). The mobile phase was 0.4% triethylamine aqueous solution, acetonitrile, and methanol (*v:v:v* = 75:10:15). Column temperature, 25 °C; flow rate, 1 mL/min; injection volume, 50 μL. The limits of detection of ENR in the solution and algae were 95.15% and 81.63%, respectively. The content of ENR was calculated by a standard curve with R^2^ > 0.99.

### 2.6. Statistical Analysis

All experiments were repeated three times. Data are given as mean ± standard deviation for each treatment and the control. One-way analysis of variance (ANOVA) was performed to determine significant differences among test concentrations using IBM SPSS Statistic 21.0.0 software (SPSS, Chicago, IL, USA). A *p* value < 0.05 was considered statistically significant. The Logistics model was fitted using Origin8.5 software (OriginLab, Northampton, MA, USA).

## 3. Results

### 3.1. The Uptake of ENR in the Algae

#### 3.1.1. Concentrations of ENR in the Culture Solution

Under exposure to ENR, the residual ENR in the culture solution gradually increased from 4.49 ± 0.12 μg/L to 298.84 ± 9.12 μg/L (Figure 1a). In the 10 μg/L exposure group, the residual ENR at 96 h was the minimum and significantly lower than that at 48 h (6.91 ± 0.08 μg/L, *p* < 0.05). The same changing trends were observed in the other exposure groups, indicating that algae were absorbing ENR from the solution during the entire growth process.

Notably, the rates of reduction in ENR concentrations in the culture solutions decreased with the increase in the exposure doses (Figure 1b), where the rates reduced sharply at the concentration of 0–150 μg/L and then remained steady at 2.38 ± 0.02%. The uptake of ENR by the algae was inhibited obviously at concentrations >150 μg/L. The control group without algae showed a reduction rate of 2.27 ± 0.04% due to the antibiotic photolysis. The maximum reduction rates were observed in the 10 μg/L exposure group (30.95 ± 0.51% and 55.01 ± 2.02%, respectively).

#### 3.1.2. Concentrations of ENR in *S. obliquus*

The concentration of ENR increased from 10.55 ± 0.13 μg/g to 18.34 ± 0.27 μg/g with the exposure concentrations, which were more significant in the 96 h treatment than in the 48 h treatment (Figure 2a). However, it was found that with the increase in the exposure concentration, the ratio of ENR concentrations in algae to those in the culture medium gradually decreased (Figure 2b), confirming that high concentrations of ENR inhibited the absorption and accumulation of algae in accordance with the above observations. Moreover, the accumulation of ENR after 96 h of exposure was significantly higher than that after 48 h in the low-concentration exposure group; however, the accumulation of ENR was almost the same during the two culture periods in the high-concentration exposure groups.

### 3.2. Effects of ENR on the Growth Inhibition of S. obliquus

#### 3.2.1. Relationship between the Algal Cell Density and the Absorbance

The curve between algal cell density (y) and OD_680_ (x) obtained by a series of dilution experiments was y = 310.8 x – 2.173 with an R^2^ of 0.998. The algal cell density was calculated according to the equation to indicate the GIR of *S. obliquus*.

#### 3.2.2. The GIR of *S. obliquus*

In general, the measured ENR showed obvious inhibitory effects on the growth of *S. obliquus*. The presence of the ENR in the culture medium significantly affected the cell division of *S. obliquus* and this negative effect was dose-dependent (*p* < 0.05). The GIR of *S. obliquus* increased from 5.31 ± 0.54%, 11.55 ± 1.61%, 8.27 ± 1.47%, and 6.77 ± 0.58% to 81.15 ± 1.54%, 90.22 ± 3.93%, 91.60 ± 3.27%, and 93.87 ± 2.08%, respectively, after treatment with 10–300 μg/L ENR for 24 h, 48 h, 72 h, and 96 h, respectively. With the elongation of the treatment duration, the inhibition rate in the relatively low-dose treatment groups (e.g., 10 μg/L) showed a downward trend. In contrast, the GIR in the higher-concentration groups (e.g., 80 μg/L to 200 μg/L) increased continuously. In the 80 μg/L treatment group, the GIR was 44.38 ± 0.44%, 53.40 ± 1.95%, 55.62 ± 1.27%, and 66.12 ± 2.05%, respectively, for 24, 48, 72, and 96 h of culture. Similar experimental results were also observed in the 120, 180, and 200 μg/L treatment groups.

#### 3.2.3. EC_50_ of ENR on *S. obliquus*

The growth inhibition curve of *S. obliquus* is shown in Figure 3. The parameters and corresponding EC_50_ values involved in the Logistics model are detailed in Appendix A. It was found that the EC_50_ value of the 24 h treatment was the largest (119.74 ± 5.56 μg/L) and almost twice as large as that of the 48 h, 72 h, and 96 h treatments. A slight fluctuation was observed in the EC_50_ values of the 48 h, 72 h, and 96 h treatments, which were ranked as follows: 72 h (64.37 ± 0.96 μg/L) > 48 h (58.09 ± 1.82 μg/L) > 96 h (52.64 ± 1.33 μg/L). Considering the results presented in Section 3.1 and Section 3.2, the exposure concentrations of 10, 50, 80, 120, and 180 μg/L were selected to conduct the subsequent experiments.

### 3.3. Effects of ENR on the Photosynthesis of S. obliquus

#### 3.3.1. The Changes in Chlorophyll Contents

The changes in Chl a, Chl b, and Total Chl contents of *S. obliquus* after 48 h and 96 h of ENR treatment, respectively, are depicted in Figure 4. The contents gradually decreased with the concentration of the ENR. After 48 h of exposure, the content of Chl a decreased to 89.61 ± 2.52%, 80.51 ± 3.84%, 69.99 ± 2.35%, 50.86 ± 1.96%, and 40.85 ± 4.23% in the 10, 50, 80, 120, and 180 μg/L experimental groups, respectively, compared with the control. The same decreasing trends were observed in the contents of Chl b and Total Chl. Remarkably, the chlorophyll content increased in the low-concentration groups after 96 h of treatment. For example, in the 10 μg/L experimental group the Chl a, Chl b, and Total Chl contents were 1.04, 1.31, and 1.18-fold those of the control, respectively. In addition, it was observed that the color of the algal solution gradually faded from green to white with the increase in the treatment concentration, and finally turned white. The chlorophyll content in the high-concentration groups decreased to a very low level, and the growth of algae was seriously inhibited.

#### 3.3.2. Changes in Chlorophyll Fluorescence Parameters

In this study, the chlorophyll fluorescence kinetic parameters and rapid fluorescence response curve of *S. obliquus* were measured to reveal the effects of ENR on the photosynthesis of algae. The values of F_0_, F_m_, F_v_/F_m_, and F_v_/F_0_ of *S. obliquus* after treatment with ENR for 48 h and 96 h are shown in Appendix A. It was found that F_0_ increased with the ENR concentration. The values were 1.42, 1.88, 2.27, 3.22, and 4.15-fold at 10, 50, 80, 120, and 180 μg/L, respectively, compared with the control, under ENR stress for 48 h.

In addition, the F_v_/F_m_ and F_v_/F_0_ values in all the treatment groups were lower than those in the control group and indicated a negative correlation with the exposure concentration. After 96 h of exposure, the F_v_/F_m_ of each group decreased to 87.43 ± 2.44%, 61.92 ± 3.08%, 59.58 ± 1.76%, and 17.41 ± 0.35% at 10, 50, 80, and 120 μg/L, respectively; and F_v_/F_0_ decreased to 70.57 ± 3.51%, 36.78 ± 2.93%, 34.06 ± 1.86%, 6.16 ± 0.38%, and 2.05 ± 1.06% at 10, 50, 80, and 120 μg/L, respectively. Moreover, the F_v_/F_m_ value recovered during the treatment with 10 μg/L ENR between the exposure times of 48 h and 96 h owing to the resistance produced in PS II of the cyanobacteria, in accordance with previous studies. However, there were signs of recovery at concentrations >10 μg/L in F_v_/F_m_.

The F_v_/F_m_ of *S. obliquus* after treatment with 10, 50, 80, 120, and 180 μg/L of ENR for 48 h and 96 h was also measured using a chlorophyll fluorometer (Figure 5) The detected color of the treated groups changed from blue or green to yellow or orange, and finally to red. With the extension of the treatment time, the orange of the algae in the 120 μg/L group deepened, and the algae in the 180 μg/L group turned red at 96 h.

The *Y* (II) of *S. obliquus* after treatment with 10, 50, 80, 120, and 180 μg/L of ENR for 48 h and 96 h is shown in Figure 6. It can be observed that *Y* (II) exhibited a significant decreasing trend with the ENR concentration under the same light intensity that was more obvious under a low light intensity. With the increase in light intensity, the *Y* (II) of *S. obliquus* gradually decreased and tended to be stable after reaching a certain light intensity. In addition, it can be observed that under the same light intensity, the *Y* (NO) in each treatment increased with the concentration of ENR. For example, under the light intensity of 80 µmol m^−2^/s, the *Y* (NO) increased by 1.03, 1.44, 1.58, 1.59, and 1.68- fold, respectively, at the concentrations of 10, 50, 80, 120, and 180 μg/L compared with the control after 48 h of treatment.

The value of NPQ decreased in a dose-effect manner for *S. obliquus* under ENR stress. After 96 h of treatment, the NPQ decreased to 38.29 ± 0.86%, 30.31 ± 1.74%, 31.75 ± 3.03%, 24.57 ± 2.75%, and 16.23 ± 0.32%, respectively, at 10, 50, 80, 120, and 180 μg/L of ENR at the light intensity of 801 µmol·m^−2^/s. A similar result was obtained for the alteration in ETR; i.e., the value decreased with the increase in the ENR concentration and the extension of time.

## 4. Discussion

In recent decades, antibiotics have been extensively and excessively used to cure human and animal diseases and received an increasing amount of attention regarding their potential harmful effects on the environment. Quinolone antibiotics are a class of synthetic antibiotics that can inhibit DNA synthesis by complexing with DNA ligases and synthases [31]. Quinolone antibiotics constituted 13% of the Chinese antibiotic market share in 2017 and have been detected in various environmental media and biological tissues [30]. However, because of their persistence and bioaccumulation, it is difficult and inefficient to remove these pollutants via engineering disposition methods, making it easy for them to be released into aquatic environments and exert toxic effects on aquatic ecosystems [32,33,34].

As a typical and widely used quinolone antibiotic, ENR, as assayed in this study, altered the growth of *S. obliquus.* The EC_50_ values obtained were close to those reported for other microalgal species in the literature. Wollenberger et al. (2000) found that the EC_50_ values of penicillin and chlortetracycline for *Microcystis* were 6 μg/L and 50 μg/L, respectively [35]. Wan et al. (1998) compared the effects of moxifloxacin stress and gatifloxacin stress on the growth of *Microcystis aeruginosa* and found that the 96 h EC_50_ values of the two antibiotics were 60.34 and 25.30 μg/L, respectively [30]. These results suggest that microalgae are strongly affected by antibiotics, despite being non-target organisms. In our study, the GIR and EC_50_ of ENR on *S. obliquus* indicated that ENR stress had an adverse impact on the growth of *S. obliquus* in a dose-dependent relationship, and the GIR was related to the exposure duration. The EC_50_ value during the 48 to 96 h period generally decreased, indicating that the algae recovered rapidly in a short time period, while the resistance of the algae to the poison gradually decreased with the extension of the treatment time. This shows that ENR at a higher concentration did not cause a fatal level of toxicity in *S. obliquus* during a short period of exposure, and the growth of *S. obliquus* recovered to a certain extent, which might be because the stress resistance mechanism was activated. However, the inhibitory effect would be irreversible when the concentration exceeds a certain range. This phenomenon implies that quinolone antibiotics could inhibit the growth of algae, reduce the number of carbon sinks, and affect the carbon cycle of the ecosystem [36,37].

Due to the complex structure and large surface area of membranes (e.g., the cell wall and the cell membrane) of algae, there is a large number of adsorption sites for organic pollutants on their surface, which is conducive to the absorption and accumulation of ENR [9,10]. The results show that the growth of *S. obliquus* increased the absorption of ENR. However, it was found that with the increase in the exposure concentration, the ratio of ENR concentrations in the algae to those in the culture medium gradually decreased from 2.69 to 0.06, confirming that a high concentration of ENR inhibited the absorption and accumulation of algae. Moreover, the accumulation of ENR after 96 h of exposure was significantly higher than that after 48 h in the low-concentration exposure group; however, the accumulation of ENR was almost the same during the two culture periods in the high-concentration groups. This might be because *S. obliquus* has a certain level of growth activity and the metabolic degradation of ENR was promoted upon exposure to a low concentration of ENR, while the growth of *S. obliquus* was inhibited and the adsorption of ENR was limited at higher doses. Moreover, it has been reported that antibiotics could complex with membrane proteins, damage organelle structures, and cause membranolysis [12,37]. Therefore, exposure to high concentrations of ENR (>150 μg/L) might damage algae severely, influencing the uptake of ENR from the solution. The maximum reduction rates observed in the 10 μg/L treatment group could be attributed to the negligible damage to algae at a low concentration of ENR, consistent with the results obtained for the GIR of *S. obliquus*. For this reason, many researchers have used algae to clean water that has been slightly contaminated with antibiotics based on their uptake and metabolic ability [7,8].

Several studies have shown that quinolone antibiotics can be absorbed by plants, cause leaf albinism, and inhibit plant photosynthesis [12,38,39]. Chlorophyll pigments are involved in a series of crucial photosynthetic reactions, including light harvesting, energy absorption, and energy conversion [40,41]. It was assumed that a disorder of the photosynthetic system in algae was the reason for the toxicity observed in our study. The results of photosynthesis-related experiments demonstrate that treatment with a low concentration of ENR can promote the chlorophyll synthesis of *S. obliquus*, which could be explained by “hormesis”; that is, low doses (below a certain threshold) are harmless to organisms and possibly even beneficial for their growth [42]. The occurrence of hormesis in plants suggests that the adaptive responses induced by low levels of stress promote growth and development and enhance the capacity for defense [43,44]. However, “hormesis” would disappear once the self-protection mechanism has been broken under high levels of stress. Several researchers found that stresses caused the expansion and rupture of chloroplasts containing photosynthetic pigments, leading to the failure of chlorophyll synthesis [45,46].

Regarding the contents of chlorophyll pigments, a significant decrease was observed in *S. obliquus* exposed to ENR (Figure 5). The photosynthetic pigment in chloroplasts is the basis for photosynthesis in plants. The photosynthesis system is highly sensitive to changes in the external environment [9]. The changes in photosynthesis caused by external environmental stresses can also lead to significant changes in chlorophyll fluorescence parameters. Furthermore, the decrease in the cellular content of pigments can be enhanced by the oxidative stress caused by these stresses [47], since it is one of the leading causes of pigment degradation in plants. In addition to the damage to the photosynthetic mechanism, the electron transfer efficiency for photosynthesis and the enzyme activity related to the dark reaction are also affected. By measuring and analyzing the fluorescence parameters, we can obtain information about the changes in photosynthesis and heat dissipation.

Although the chlorophyll contents were inhibited in cultured cells, an increase in the chlorophyll fluorescence was detected in this study. Under normal physiological conditions, the values of F_v_/F_m_ and F_v_/F_0_ in plants are very stable, but they will fluctuate greatly under stress conditions [48]. F_m_ showed an upward trend. This increase could be explained by a blockage of the electron transport chain in the PS II system [30], indicating an inhibitory effect localized to the oxidant side, probably due to the inactivation of some PS II reaction centers. Similar results were obtained in our previous study, in which the growth of *S. obliquus* was inhibited upon treatment with alkyl-methylimidazolium nitrate ionic liquids [30]. In addition, under the stress of a high concentration of ENR, the decline in F_v_/F_0_ was more rapid than that in F_v_/F_m_, indicating that the effect of ENR stress on the energy transfer in the algae was more obvious than that on the light energy conversion efficiency. In addition, F_v_/F_m_ recovered upon treatment with 10 μg/L ENR between the exposure times of 48 h and 96 h owing to the resistance produced in the PS II system of cyanobacteria in accordance with previous studies. However, the observed significant decrease in F_v_/F_m_ with no signs of recovery at concentrations >10 μg/L indicated that PS II suffered from irreversible damage after exposure to ENR and that this damage can contribute to the growth inhibition in *S. obliquus* [38].

*Y* (II) represents the actual photosynthetic conversion efficiency of PS II; that is, the conversion efficiency of light quanta actually absorbed by the PS II reaction center when it is partially closed [19]. Therefore, the change in *Y* (II) directly reflects the strength of the photosynthesis and the sensitivity of the algae to stress, effectively reflecting the actual [27] photosynthetic efficiency of the photosynthetic system. The change in *Y* (II) in our study indicated that the variation in the actual light energy conversion efficiency of PS II, which might be related to the damage to *S. obliquus* and the fixation and assimilation processes of photosynthetic carbon, was affected. This observation partly explains why the chlorophyll contents were inhibited while the fluorescence was increased.

*Y* (NO) represents the excitation energy absorbed by PS II; that is, the energy that passively dissipates to heat and fluorescence [12]. It is commonly used to characterize light damage to plants. The higher *Y* (NO) value detected in *S. obliquus* indicated that the light energy conversion efficiency and the self-protection mechanism of the algae were weakened or damaged and that the excess light energy could not be completely consumed. In this case, when the light intensity exceeds the threshold of the algal cells, the continuous stream of light will lead to continuous damage to the algae [5].

The NPQ is often used to measure the amount of light energy that dissipates to excitation energy in the form of heat and reflects the ability of plants to protect against light damage [9]. As a self-protection mechanism in photosynthesis, the NPQ can convert the excess light energy to heat and protect the PS II system. The reason why the NPQ decreased with the increase in the treatment concentration was that ENR stress damaged the PS II system of *S. obliquus* and it gradually lost the ability to dissipate excess heat energy. In addition, the ETR represents the apparent electron transport efficiency of PS II under the actual light intensity [24]. It can be observed that the ETR decreased with the increase in the ENR concentration and the extension of time, in line with the results on the change in the chlorophyll contents. 

In conclusion, the obtained results show that ENR provoked adverse effects on the growth of *S. obliquus*, damaged the photosystem of *S. obliquus* microalgae, and affected the carbon storage in aquatic environments.

## 5. Conclusions

ENR at low concentrations (~10 μg/L) could enhance the growth of *S. obliquus* due to hormesis, reflected by the promotion of photosynthesis. Relatively high doses of ENR inhibited the normal growth of *S. obliquus*, and the elongation of the cultivation duration relieved the toxicity to some extent. The Chl contents were negatively correlated to the treatment concentrations. Fluorescence parameters were significantly influenced by ENR, indicating that ENR might damage the photosystem and hinder the normal synthesis of photosynthetic pigments. These parameters also suggest that ENR affected the carbon fixation and assimilation processes of *S. obliquus* by damaging its photosystem. The toxicity assessments performed in this study indicate that ENR poses a considerable risk to aquatic environments and provide new insights into the mechanism by which quinolone antibiotics reduce the carbon sink in aquatic systems.

## Figures and Tables

**Figure 1 ijerph-19-05545-f001:**
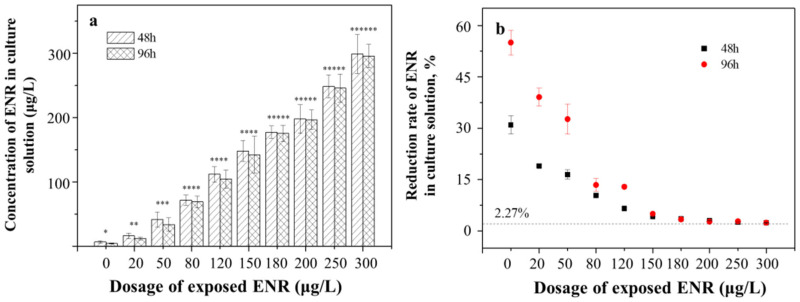
The residual (**a**) and rate of reduction in (**b**) ENR concentrations in the algae culture solution under different exposure dosages at 48 h and 96 h. The number of “*” was referred as the significant between-group variance. The same number means no significant difference, whereas the different number means the significant difference.

**Figure 2 ijerph-19-05545-f002:**
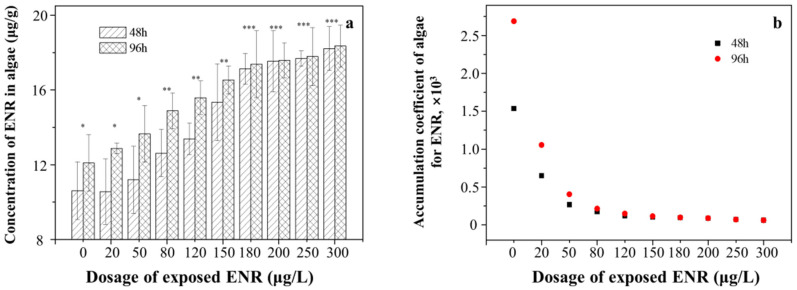
The accumulation of ENR in algae under different exposure dosages at 48 h and 96 h. (**a**) The amount of accumulated ENR; (**b**) the accumulation coefficient of ENR. The number of “*” was referred as the significant between-group variance. The same number means no significant difference, whereas the different number means the significant difference.

**Figure 3 ijerph-19-05545-f003:**
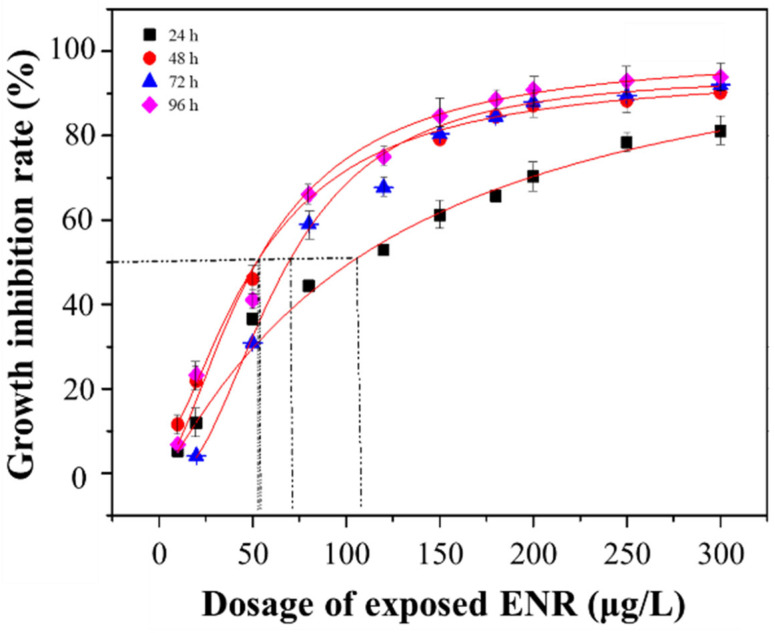
The growth inhibition rate of algae under different exposure dosages of ENR.

**Figure 4 ijerph-19-05545-f004:**
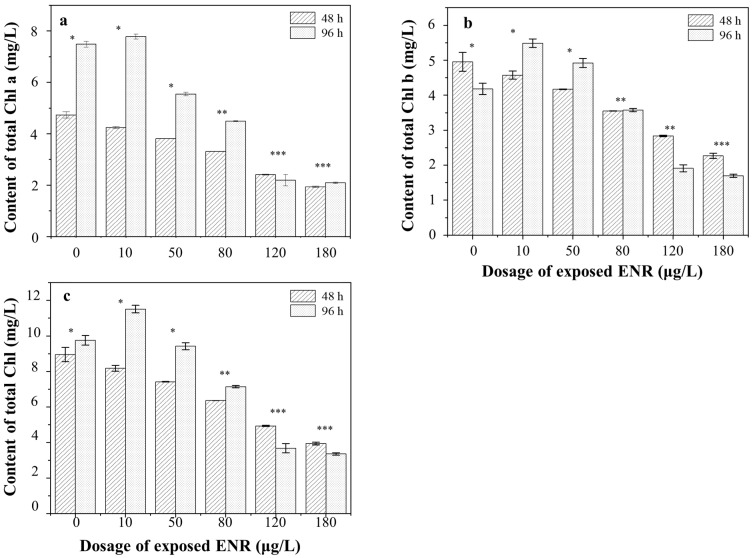
The chlorophyll contents in algae under different exposure dosages of ENR. (**a**) Chl a content; (**b**) Chl b content; (**c**) Total Chl content. The number of “*” was referred as the significant between-group variance. The same number means no significant difference, whereas the different number means the significant difference.

**Figure 5 ijerph-19-05545-f005:**
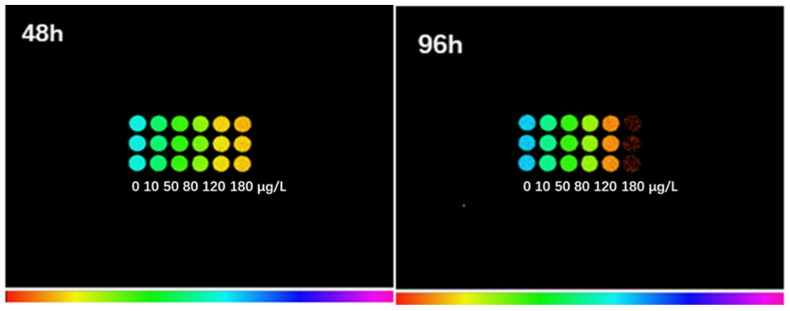
Fluorescence images of chlorophyll in *S. obliquus* after exposure to ENR for 48 h and 96 h. The colors indicate the degree of algal damage; the redder the color, the more serious the damage.

**Figure 6 ijerph-19-05545-f006:**
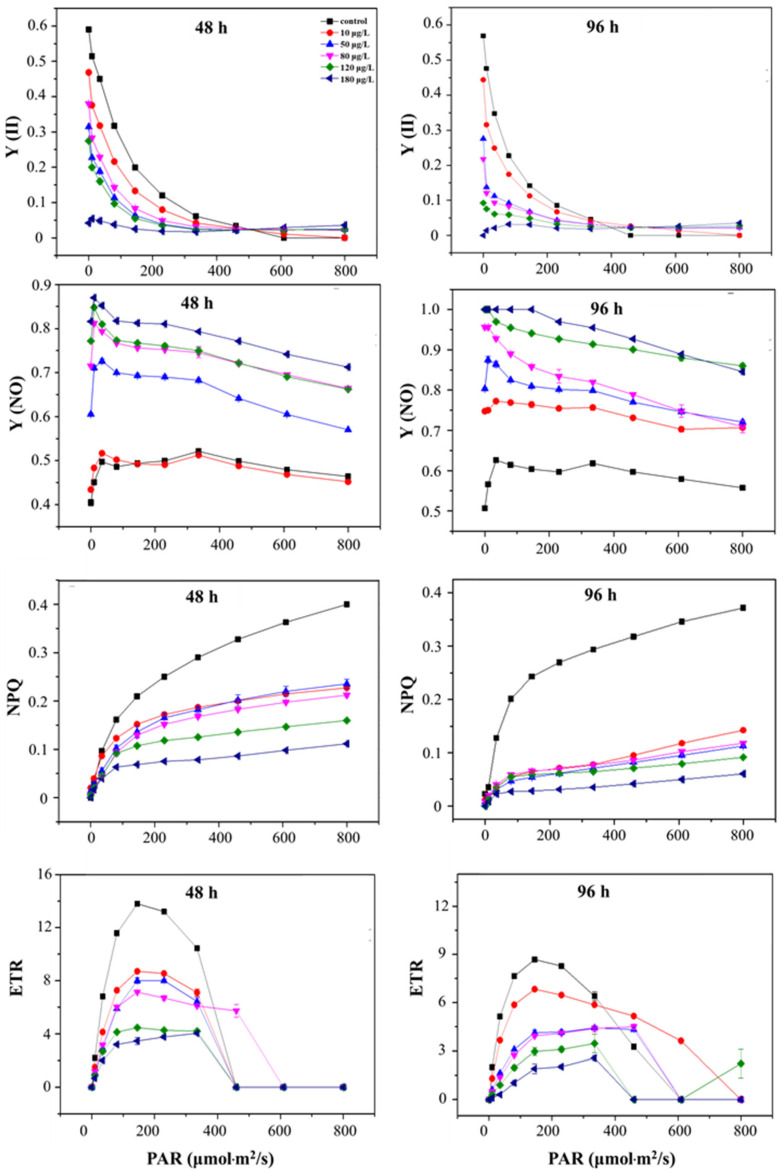
The change in chlorophyll parameters of *S. obliquus* with the dosages of 0, 10, 50, 80, 120, and 180 μg/L. PAR, photosynthetic active radiation; *Y* (II), actual photosynthetic efficiency; NPQ, non-photochemical quenching coefficient; *Y* (NO), quantum yield of non-regulatory energy dissipation; ETR, relative electron transfer efficiency.

## Data Availability

Not applicable.

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
