# Peer review of "Photosynthetic Toxicity of Enrofloxacin on Scenedesmus obliquus in an Aquatic Environment"

_ijerph, 2022, doi:10.3390/ijerph19095545_

Round 1

Reviewer 1 Report

Aquaculture facilities are a potential source of antibiotics to aquatic environments, posing adverse effects on the human health. The authors evaluated the toxicity induced by enrofloxacin (ENR) on the algae (Scenedesmus obliquus). The results indicated that ENR poses a considerable risk to aquatic environments, affects the carbon sinks and even has adverse effect on human health.This work is practical significance and meaningful.

General comments: The authors evaluated the photosynthetic toxicity induced by enrofloxacin on the Scenedesmus obliquus, and found enrofloxacin caused a disorder in the electron transport of photosynthesis of algae, and the carbon fixation and assimilation process were also damaged. My concern is that the photosynthesis of algae is closely related to the defense mechanism of plants and the authors also figured out that the damaged tissues might be caused by the perturbed antioxidant. However, the related parameters, like SOD, POD or APX enzymes were not determined. Generally, the analysis of chlorophyll contents and fluorescence provided intuitive and reliable evidence for the inhibition of antibiotics on algae, which broadens our horizon on the effects of environmental pollutants on algal growth and carbon fixation in the aquatic ecosystem, especially under the background of the carbon neutral. I recommend this manuscript to be accepted in Int. J. Environ. Res. Public Health after minor revision.

Specific comments:

  1. Line 61: revise the O2.
  2. Fig. 1 and Fig. 2 missed the data of control group, please add them.
  3. The description of x axis in Fig. 3 is missing.
  4. The description of y axis in Fig. 4 should be checked because there are two “chl b”; and the x axis should be stay the same with other figures.
  5. The size of Fig. 5 is too large, please make it suitable to read.
  6. Line 475: EC50 should be corrected.
  7. Reference of 22, the name of the journal should be abbreviated.
  8. The font in letters in figures should be enlarged. It was too small to see. Generally, the quality of photos in the figures should be improved.

Author Response

Response to Reviewer 1 Comments

Reviewer 2 Report

The authors evaluated the toxicity induced by enrofloxacin (ENR) on the algae (Scenedesmus obliquus) and analyzed the uptake of ENR and the change on the growth and photosynthesis of algae. This is a meaningful and practical study. The paper can be considered for the publication subject to answers to the following comments.

The results should be statistically significant and noted in the figure.

The detail of some methods should be given. For example Enrofloxacin is insoluble in water, how did the authors deal with it in this paper.

Line 235: The sentence is not clear, probably a part is missing for the algae.co

Line 246: The “ENR assayed” should be revised as the “measured ENR”.

Line 432: “Thus” in this sentence should be removed for the reading.

Line 442: “Taken into account the obtained results” could be better to be replace with “In conclusion”.

References 7, 15, 22, 25 need to be improved.

Author Response

Response to Reviewer 2 Comments

Reviewer 3 Report

General comment:

The topic has international appeal. The goal of this research was to calibrate a quasi related experiment displayed model. The article is really and contains a key management message. The introduction provides a good, generic outline that immediately gives an impression of the vast range of applications of studies focusing on mildly antibiotics-contaminated reactions, and should spark the readers' curiosity. It is an exciting experience to conduct such a broad field trial in an aquatic environment.

Specific comments:

The following are the paper's major problems and weaknesses. Please find the following remarks for your consideration:

  1. The research area part has to be clarified because it appears that only a small area was sampled and there is no actual replication (i.e., growth inhibition rate and EC5). Please explain how the samples have been collected. In addition, more citations are desired to support the comments in the Result and Discussion sections. The authors referenced a variety of theories that were unrelated to the study's goal. The thoughts in the Discussion section should be more developed and relevant to the study's goal.

  2. The criteria used to classify and measure the number of shuffling complicated approaches for effectiveness are unclear since organisms can be split by trait, but no ecological division has been identified.

  3. This study makes an admirable effort to apply data techniques to a large volume of data generated by a variety of algae adaptations controlled on entirely different grounds. Even though a few popular ecological concepts regarding the fatal toxicity of S. obliquus may be similar in these eco-structures. Is it a necessity, for example, to examine growth patterns in certain boom interests and the metabolic degradation of ENR? An evaluation of a dynamic increase version that is multiannual requires a different method and is therefore managed on entirely different grounds.

  4. Please clarify which data are included in the concentrations of chlorophyll in algae when exposed to various ENR doses. The authors must justify the use of a detailed comparison of selecting a model to anticipate the commencement of growth in the study's purpose.

Constructive feedback

Once the data analyses have been repeated, the conclusion section might have to be rewritten. Instead of referring to specific sites, I recommend focusing on the main consequences of the recent decline in chlorophyll fluorescence data. The conclusion section should be updated to focus on the study's key objectives and to be consistent with the results collected. Yet, they did not examine the long-term effects of changes in chlorophyll properties (and no past studies on this topic are acknowledged in the Discussion). The conclusion should, in general, include the following: 'discussion of the results and not just repetition of the results'.

I recommend reiterating the analyzed data since (1) the persistence of chlorophyll fluorescence is ambiguous and sometimes combined; (2) the error distribution of samples is used in immediate adjustment to test the effects of environmental variables on count data, i.e., the number of all traits, that should be evaluated using negative binomial and Poisson error distribution samples, respectively; (3) the accumulation of ENR in algae under different exposure dosages.

Methodological rigidity in presenting its experiential research:

Despite the good quality of the methods, the mathematics is not properly reworded because conclusions are not supported by the data. Additionally, the results are not well presented; they ought to be specific and relevant to the main aim of the study. For biophysical parameters, machine learning or regression algorithms would probably need to be applied and revised. The ANOVA is insufficient for each treatment and the control. More advanced methods are needed. It is quite standard to model toxic effects at multiple scales and is appropriate for a study of that kind.

Summary:

The manuscript aims to investigate robustness and uncertainty using an experimental approach. Exposure to excessive quinolone antibiotics, enrofloxacin (ENR), as described here is nothing new, or surprising, and has been shown before in numerous other studies. If the major concerns above will be addressed satisfactorily, I believe the document can be a source of scientifically novel information. With no way regret, I must inform you that, based on the advice received, I am able to accept your manuscript for publication. However, for now, I am unable to recommend this work for publication in this journal. Major revisions are necessary before it is suitable for publication.

Author Response

Response to Reviewer 3 Comments

Reviewer 4 Report

The aim of the manuscript is to evaluate the toxicity of an antibiotic the enrofloxacin (ENR) on Scenedesmus obliquus in aquatic environment. The topic of this manuscript is interesting, and it contains a great amount of interesting data. However, it needs some edits both in the form and substance to improve its quality and some points deserve to be a little more discussed by the authors in order to improve the quality of the manuscript before publication.

Major comments:

  • Only one antibiotic the enrofloxacin (ENR) was used in this study, therefore in the title the term antibiotics was not adequate. It’s better to indicate the toxicity of an antibiotic the enrofloxacin……
  • It is clearer to include each figure where it is described in the text and not to collect all the figures in a single section.
  • In the materials and methods section, it is not indicated how many times each experiment was repeated? If more than one time, the results should be expressed in mean with standard deviation.
  • In the section 2.2, the authors indicated that data were given as mean ± standard deviation for each treatment and the control. The one-way analysis of variance (ANOVA) was used to determine significant differences. However, in some figures (3 and 6), data are not given as mean ± standard deviation and no symbol (asterisk for example) was indicated for significance difference between data in all figures.-        Lines 238-239: the EC50 value of treatments for 48 h, 72 h and 96 h, which could be ranked 238 as: 72 h (64.37 μg/L) > 48 h (58.09 μg/L) > 96 h (52.64 μg/L). It is strange that the EC50 value at 72 h was higher than that at 48 h. In the figure 3, the curves for 48 h and 96 h are superimposed.
  • The conclusion needs to be shortened and focus only on the main results obtained.    

Minor comments:

  • Line 11, adverse effects on the human health and ecosystems. Because this study is focused on the effect of the antibiotic on an alga species in the aquatic environment and not on the human health.
  • Lines 14-15: levels of ENR in algae were 10.61–18.22 μg/g and 12.09–18.34 μg/g after 48 h and 96 h treatment, respectively.
  • Line 15: with the concentration for 50% of maximal effect. The value is for EC50 (concentration given 50% effect and not a maximal effect). The sentence should be revised and checked throughout the manuscript.
  • Line 49: Wei et al. year should be added, Wei et al. [year].
  • Line 60: should be O2 and not O2.
  • Line 127: 4,000 rpm, after (for example, line 162) the centrifugation speed is given in g. It should be better to harmonize throughout the manuscript and give all in g.
  • Line 332: shloud be Wollenberger et al. [year].
  • Line 333: Microcystis instead of Microcystins.
  • Line 333: were 0.006 mg/L and 0.05 mg/L, the values given for the antibiotic in this study are expressed in µg/L. It is better to give all value in µg/L to facility of comparison between species.
  • Line 334: Wan et al. [year].
  • Lines 336-337: These results suggested that microalgae are strongly affected by 336 antibiotics, despite being non-target organisms. Microcystis is a prokaryote cyanobacterium, therefore it is normal that it was more sensitive to antibiotic like bacteria. However, the alga species used in this study is a eukaryote microorganism and its sensitivity against antibiotics was different. This should be more discussed.
  • Line 403: Wan et al. [year].

Author Response

Response to Reviewer 4 Comments

Round 2

Reviewer 3 Report

An easily readable and well-understood methodology is presented.
The methodology is better in any case. The errors have been corrected. The manuscript is free of errors and inaccuracies. Thanks for considering my suggestions.